REGISTERED REPORT PROTOCOL

# Application of concentrated growth factor in mandibular third molar extraction: A protocol for systematic review and meta-analysis

**Hengxiao Zhang**[☯], **Jianyong Dong**[ID][☯], **Xiaoliang Wang, Xiaodong Sun, Jin Wang**[ID]*

Gaoxin Branch, Jinan Stomatological Hospital, Jinan, Shandong Province, China

☯ These authors contributed equally to this work.
* 634833675@qq.com

## Abstract

### Objective

We will perform the systematic review to evaluate the effect of applying concentrated growth factor (CGF) on relieving postoperative complications and promoting wound healing following mandibular third molar extraction.

### Methods

The PubMed, Web of Science, Embase, Cochrane Library, China National Knowledge Infrastructure (CNKI), Wanfang Database, China Biology Medicine Disc (CBM), and VIP Databases will be comprehensively searched up to May 31, 2024. Randomized controlled trials (RCTs) examining the application of CGF after mandibular third molar extraction will be included. The protocol was registered in PROSPERO, and the registration ID was CRD42023463234. Two reviewers will conduct the literature search, eligible study selection, data extraction, and bias risk assessment (using the Cochrane Risk of Bias 2.0 tool). Data analysis will be performed with RevMan software (version 5.4).

### Results

The results of this study will be available in a peer-reviewed journal.

### Conclusion

Our study will provide scientific evidence regarding the efficacy of applying CGF in mandibular third molar extraction.

## Introduction

The extraction of the mandibular third molar is a common oral and maxillofacial procedure. Following the procedure, various types of postoperative complications occur, including pain,

**Data Availability Statement:** No datasets were generated or analysed during the current study. All relevant data from this study will be made available upon study completion.

**Funding:** This work was supported by the Science and Technology Planning Project of Jinan Health Commission (2021-1-44). There was no additional external funding received for this study.

**Competing interests:** The authors have declared that no competing interests exist.

swelling, and trismus [1]. In addition to short-term postoperative adverse events, the extraction of mandibular third molars lead to long-term damage to the periodontal tissue around second molars, which cannot be ignored [2]. The extraction of an impacted third molar can result in a residual alveolar defect, the loss of attachment, and periodontal pocket formation at distal sites of second molars [3]. However, challenges remain with respect to relieving patient discomfort, accelerating wound healing and promoting tissue regeneration.

Concentrated growth factor (CGF), a third-generation platelet concentrate [4], has many advantages compared with first- and second-generation platelet concentrates, i.e., platelet-rich plasma (PRP) and platelet-rich fibrin (PRF) [5]. The growth factors and CD34$^+$ stem cells in CGF increase its ability to promote bone formation and tissue healing. CGF has a harder, larger and denser fibrin structure and serves as a scaffold at the application site, thus promoting tissue regeneration [6]. It has been widely used in oral and maxillofacial procedures, such as oral implants, maxillary sinus floor elevation, periodontal tissue regeneration, alveolar ridge preservation and tooth extraction [7–9]. CGF has been demonstrated to be beneficial for decreasing the incidence of postoperative complications [10], healing soft tissue [11] and promoting ridge preservation [12] after third molar surgery. New studies about the promising efficacy of CGF in third molar extraction are constantly emerging. However, there are some inconclusive results. Previous research has indicated that CGF has no additional beneficial effects on pain, swelling, or trismus after surgery [13]. Another study revealed that CGF had no positive effects on the distal periodontal depth or bone regeneration of the second molar [14]. However, these previous studies had small sample sizes, insufficient analysis factors, methodological differences and shorter follow-up periods, thus preventing us from drawing firm conclusions.

To our knowledge, given the small number of relevant randomized controlled trials (RCTs), no systematic review has been conducted on this specific topic. There is limited scientific evidence supporting the widespread clinical application of CGF in mandibular third molar extraction. Because more articles have recently been published on this subject, we conduct this systematic review. The objectives of the current review are as follows: 1) to determine whether CGF relieves postoperative clinical outcomes (pain, swelling, and trismus) in the short term; 2) to evaluate whether CGF promotes the healing of soft tissue; and 3) to determine whether CGF promotes periodontal regeneration (soft tissue and bone healing) in the distal area of the adjacent second molar following mandibular third molar surgery.

## Methods

This systematic review will be conducted in accordance with the PRISMA guidelines [15]. The PRISMA-P checklist is included in S1 Checklist. The protocol has been registered in PROSPERO (CRD42023463234). The PICOS strategy was used to construct the following research parameters:

- population (P): patients who underwent mandibular third molar extraction;

- intervention (I) and comparison (C): in the intervention group, CGF was placed into the tooth socket, while no treatment was used in the control group;

- outcomes (O): the primary outcome and secondary outcome are described below;

- study design (S): RCTs including human subjects.

The research question is as follows: Does the application of CGF relieve postoperative clinical outcomes, improve the healing of soft tissue and promote regeneration of periodontia after mandibular third molar extraction?

## Information sources and search strategy

Two independent reviewers (Hengxiao Zhang and Jianyong Dong) will search the PubMed, Web of Science, Embase, Cochrane Library, China National Knowledge Infrastructure (CNKI), Wanfang Database, China Biology Medicine Disc (CBM), and VIP Databases to identify potentially relevant articles. The ProQuest, Google Scholar, and Open Grey will be searched for grey literature. We will resolve disagreements by consulting a third reviewer (Jin Wang). The search strategy will be developed by a combination of medical subject terms (MeSH) or EMBASE Subject Headings (Emtree) terms along with the corresponding free text words. The free text words will include "concentrated growth factor" or "CGF", and "third molar". RCTs that included human subjects and were published up to May 31, 2024 will be considered. The search strategy for PubMed is presented in Table 1, and the detailed search strategy for other databases is available in the S1 Appendix.

## Study selection

Two independent reviewers (Hengxiao Zhang and Jianyong Dong) will screen the studies. We will import all of the retrieved studies into Endnote software (version X9), and duplicate articles will be automatically excluded. After screening the title and abstract, we will obtain the full text of the relevant research. The reviewers will assess whether each article meets the inclusion criteria to determine its eligibility. We will also manually search the reference lists of the included articles to identify additional relevant studies. If there is any disagreement, it will be decided by the third reviewer (Jin Wang). The main reasons for the exclusion of articles and the research selection process will be documented. The PRISMA flow chart is shown in Fig 1.

## Inclusion and exclusion criteria

The inclusion criteria will be as follows: 1) randomized clinical trials, 2) studies evaluating the use of CGF in mandibular third molar extraction, 3) studies examining pain, swelling, trismus, or soft tissue healing as primary or secondary outcomes, and 4) studies mainly published in Chinese or English. The relevant exclusion criteria will be as follows: 1) retrospective studies, 2) studies using CGF along with other elements, and 3) studies for which relevant data were not reported.

## Primary outcome and secondary outcome

The removal of mandibular third molars is often accompanied by unavoidable adverse effects, such as pain, swelling, and trismus. The primary outcome will be postoperative pain. The secondary outcomes will include facial swelling, trismus, soft tissue healing, bone height and periodontal parameters (i.e., probing depth, gingival recession and clinical attachment level).

**Table 1. Details of the search strategy for PubMed.**

| No | Search item |
|----|-------------|
| #1 | (Concentrated growth factor) OR (CGF) OR (Autologous platelet concentrate) OR (Blood platelets)) OR (Platelet concentrates) OR (Intercellular Signaling Peptides and Proteins [MeSH Terms]) OR (Blood Platelets [MeSH Terms]) OR (Blood Proteins [MeSH Terms]) OR (Fibrin [MeSH Terms]) |
| #2 | (Tooth, Impacted [MeSH Terms]) OR (Molar, Third [MeSH Terms]) OR (Molar [MeSH Terms]) OR (Third molar) OR (Impacted tooth)) OR (Wisdom tooth) |
| #3 | (Tooth Extraction [MeSH Terms]) OR (Tooth Socket [MeSH Terms]) OR (Dental extraction) OR (Tooth extraction) OR (Dental surgery) OR (Tooth Socket) |
| #4 | (randomized controlled trial) OR (randomized controlled trials as topic [MeSH Terms]) OR (randomized controlled trial [Publication Type]) |
| #5 | #2 OR #3 |
| #6 | #1 AND #4 AND #5 |

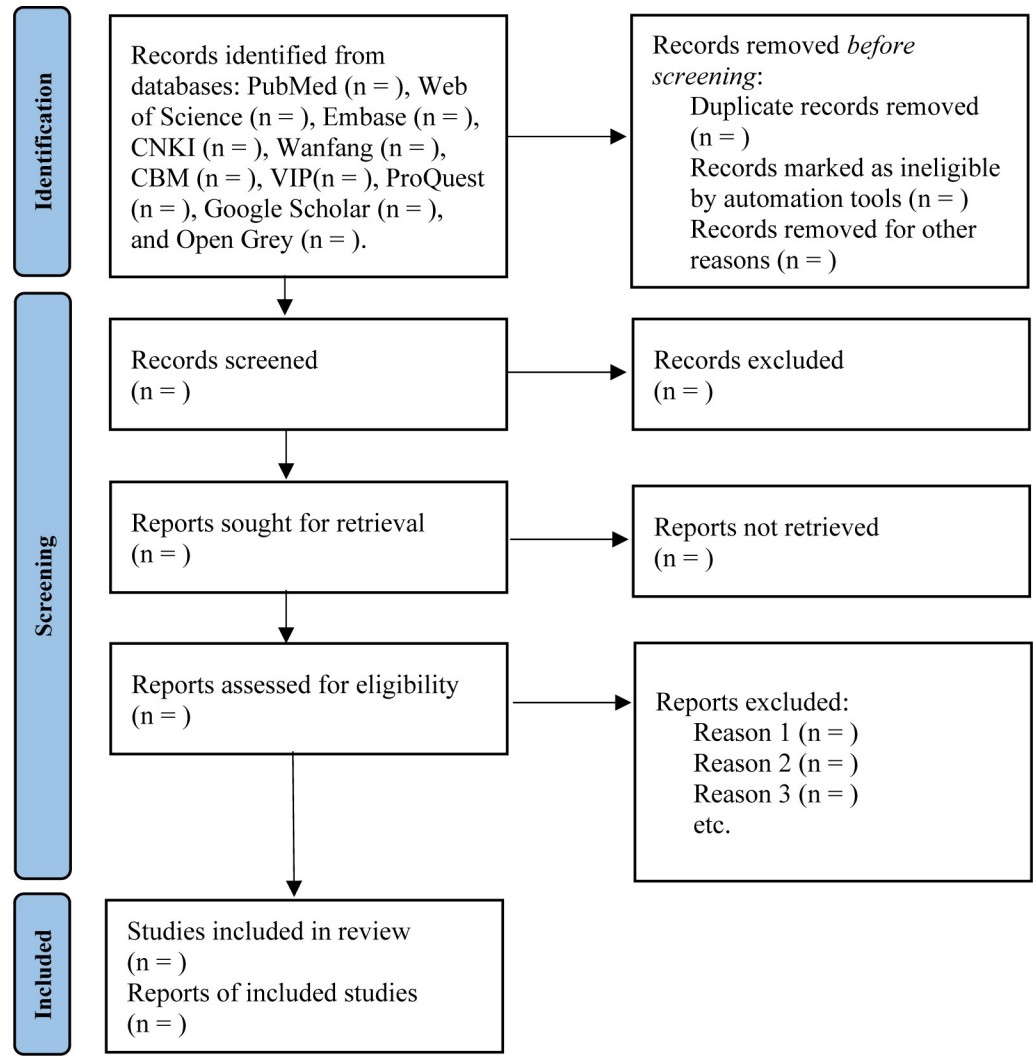

**Fig 1. Flow chart and descriptions of study selection.**

## Data extraction

A previously prepared data extraction form will be used. The details will be extracted: authors, year, study design, participant age, male-to-female ratio, number of patients, interventions (CGF gel vs. CGF membranes), flap design (triangular flap vs. envelope flap), bone removal (yes or not), operation time, primary and secondary outcomes, specific measurement methods, and duration of follow-up. Two reviewers (Hengxiao Zhang and Jianyong Dong) will extract the data in accordance with the inclusion criteria. In cases of disagreement, the third author will resolve the inconsistencies. If important data are missing, we will contact the corresponding authors.

## Risk of bias assessment

Two independent reviewers will assess the risk of bias using the Cochrane Risk of Bias (RoB) 2.0 tool, a tool used to assess the risk of bias in randomized trials [16]. The following domains of bias will be assessed: a) randomization process, b) deviations from intended interventions, c) missing outcome data, d) measurement of the outcome, and e) selection of the reported

result [17]. For each domain, the risk of bias will be graded as "high", "some concerns", or "low".

## Assessment of heterogeneity and data synthesis

All meta-analyses will be conducted using RevMan software (version 5.4). Calculation of weighted mean difference (WMD) and 95% confidence intervals (CIs) will be necessary for continuous data. In case of data are presented as medians and quartiles, we will convert them to means and standard deviations (SDs) [18]. For dichotomous variables, we will calculate risk ratios (RRs) or odds ratios (ORs) along with 95% CIs. Cochrane's Q test and the $I^2$ statistic will be used to assess heterogeneity of the studies [19]. If significant heterogeneity ($P < 0.05$ and $I^2 > 50\%$) is observed, the random effects model will be adopted, which can provide a more conservative estimate of the difference. In the cases of low heterogeneity, we will employ the fixed effect model to assure the statistical efficiency. To account for methodological heterogeneity in the measurements, a descriptive synthesis will be performed if variables could not be included in the meta-analysis.

## Subgroup analysis

If significant heterogeneity is observed, we will conduct subgroup analysis based on sex, position of the mandibular third molar, flap design, bone removal, duration of surgery and follow-up duration.

## Sensitivity analysis

We will assess the stability and reliability of the results through sensitivity analysis. We will eliminate one study at a time to determine if the study account for significant heterogeneity.

## Publication bias

If we include 10 or more articles, we will assess publication bias by using Begg's and Egger's tests [20]. The results will be presented by a funnel plot. An asymmetry in the funnel plot will be considered to indicate potential publication bias.

## Grading the quality of evidence

We will use the Grading of Recommendations Assessment, Development and Evaluation (GRADE) framework to assess the quality of evidence [21]. The quality of evidence will be categorized into 4 levels: very low, low, moderate, or high.

# Discussion

This systematic review and meta-analysis of published RCTs will be performed to evaluate the application of CGF for mandibular third molar extraction. This review could help clinicians better understand the potential advantages of CGF in third molar extraction. Many studies have proven the effectiveness of CGF in the extraction of mandibular third molar, including randomized controlled trials and retrospective clinical studies. However, there are still some controversies about the benefits of CGF. Most related studies have focused on short-term improvements in postoperative symptoms [11]. However, few relevant studies have explored the roles of CGF in promoting of periodontal tissue regeneration of third molars and adjacent second molars [12]. Our study will review the currently available evidence regarding the use of CGF in mandibular third molar extraction. This review has several limitations. Differences in surgical protocols and the outcome measurements may lead to bias in the results. This review

will only include English and Chinese studies, which may decrease the generalizability of the review. Our review will draw comprehensive quantitative and qualitative conclusions based on the currently available evidence. The results of the study will provide evidence-based medical guidance for clinicians regarding the use of CGF. Furthermore, these findings will illustrate the weaknesses of current studies and potentially inform future research.

## Supporting information

**S1 Checklist. PRISMA-P (Preferred Reporting Items for Systematic review and Meta-Analysis Protocols) 2015 checklist.**
(DOC)

**S1 Appendix. The search strategy for databases.**
(DOCX)

## Author Contributions

**Conceptualization:** Jin Wang.

**Data curation:** Hengxiao Zhang, Jianyong Dong, Xiaoliang Wang.

**Formal analysis:** Hengxiao Zhang, Xiaoliang Wang, Xiaodong Sun.

**Funding acquisition:** Jianyong Dong.

**Investigation:** Jianyong Dong, Jin Wang.

**Methodology:** Hengxiao Zhang, Jianyong Dong, Xiaoliang Wang, Jin Wang.

**Project administration:** Jianyong Dong.

**Resources:** Hengxiao Zhang, Jianyong Dong, Xiaoliang Wang, Xiaodong Sun.

**Software:** Hengxiao Zhang, Xiaoliang Wang, Xiaodong Sun.

**Supervision:** Jin Wang.

**Validation:** Jin Wang.

**Visualization:** Jin Wang.

**Writing – original draft:** Hengxiao Zhang, Jianyong Dong.

**Writing – review & editing:** Hengxiao Zhang, Jianyong Dong, Jin Wang.

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
