## [Decision Letter · Decision Letter 0]

8 Feb 2024

PONE-D-24-01317Application of Concentrated Growth Factor in Mandibular Third Molar Extraction：A Protocol for Systematic Review and Meta-AnalysisPLOS ONE

Dear Dr. Wang,

Thank you for submitting your manuscript to PLOS ONE. After careful consideration, we feel that it has merit but does not fully meet PLOS ONE’s publication criteria as it currently stands. Therefore, we invite you to submit a revised version of the manuscript that addresses the points raised during the review process. Please, provide an extensive review with any necessary updates suggested by the reviewers. 

We look forward to receiving your revised manuscript.

Kind regards,

Endi Lanza Galvão

Academic Editor

PLOS ONE

Journal Requirements:

2. In your cover letter, please confirm that the research you have described in your manuscript, including participant recruitment, data collection, modification, or processing, has not started and will not start until after your paper has been accepted to the journal (assuming data need to be collected or participants recruited specifically for your study). In order to proceed with your submission, you must provide confirmation.

"This work was supported by the Science and Technology Planning Project of Jinan Health Commission (2021-1-44)."

Reviewers' comments:

Reviewer's Responses to Questions

**Comments to the Author**

1. Does the manuscript provide a valid rationale for the proposed study, with clearly identified and justified research questions?

Reviewer #1: Partly

Reviewer #2: Yes

2. Is the protocol technically sound and planned in a manner that will lead to a meaningful outcome and allow testing the stated hypotheses?

Reviewer #1: Partly

Reviewer #2: Yes

3. Is the methodology feasible and described in sufficient detail to allow the work to be replicable?

Reviewer #1: Yes

Reviewer #2: Yes

4. Have the authors described where all data underlying the findings will be made available when the study is complete?

Reviewer #1: Yes

Reviewer #2: Yes

5. Is the manuscript presented in an intelligible fashion and written in standard English?

Reviewer #1: Yes

Reviewer #2: Yes

6. Review Comments to the Author

You may also provide optional suggestions and comments to authors that they might find helpful in planning their study.

Reviewer #1: Dear authors,

Congratulations by systematic review protocol. The topic is interesting and current. The introduction is well written, but it suggests that there are few studies on the topic and so the authors expanded the outcome. In this situation, the systematic review may not have the expected results due to this limitation and the studies might have several heterogeneity. The methodology needs some updates with newer tools in review development.

Some suggestions are made throughout the attached file.

Good luck

Reviewer #2: For the authors

General comment:

Congratulations on your work, which is very well written and outlined. After making the corrections described below, the article will be suitable for acceptance.

Specific comments:

1-Abstract:

1.1 Please insert the phrase "We performed the systematic review..." in the future, as the authors have not yet finalized the review and this article is a protocol;

1.2 Were searches carried out in the gray literature, such as Google Scholar, ProQuest, OpenGray? Has the reference list of the included articles been checked?

2-Introduction:

2.1 I think the introduction is too long. I recommend shortening the first paragraph a little;

2.2 In the sentence "Because more articles have recently been published on this subject, we conducted this meta-analysis", change the term meta-analysis to systematic review.

3-Methods:

3.1 The way the PICO strategy was described was confusing. I suggest a clearer standardization;

3.2 Was there a date limitation when conducting the search?

3.3 I suggest that the authors use the version of PRISMA in which they have the option of adding the results of the gray literature. I believe that more studies could be found and included in the review;

3.4 As one of the aims of publishing a systematic review protocol is to allow other researchers to access it and be guided in the development of their protocols, I believe that adding the search strategy for other databases, in addition to PUBMED, is relevant, even as a supplementary file;

4- Discussion:

4.1 In this sentence: "Many studies have proven the efficacy of FBC in the extraction of lower third molars; however, there is no consensus", make it clear whether these are RCTs or another study design;

7. PLOS authors have the option to publish the peer review history of their article (what does this mean?). If published, this will include your full peer review and any attached files.

Reviewer #1: No

Reviewer #2: No

---

## [Author Response · Author response to Decision Letter 0]

13 Mar 2024

List of Responses

Dear editors and reviewers:

Thank you for your comments concerning our manuscript entitled "Application of Concentrated Growth Factor in Mandibular Third Molar Extraction: A Protocol for Systematic Review and Meta-Analysis" (PONE-D-24-01317).

These comments were all valuable for improving the paper and guiding our future research. We have made corrections that we hope will meet with your approval. The revised areas and main corrections are marked in red in the revised paper. Our research will not start until the paper is accepted by the journal. 

Thank you again for your time and consideration.

Journal Requirements:

Response: We have ensured that the manuscript adheres to the style requirements of PLOS ONE.

2. In your cover letter, please confirm that the research you have described in your manuscript, including participant recruitment, data collection, modification, or processing, has not started and will not start until after your paper has been accepted to the journal (assuming data need to be collected or participants recruited specifically for your study). In order to proceed with your submission, you must provide confirmation.

Response: Our research will not start until the paper is accepted by the journal. 

"This work was supported by the Science and Technology Planning Project of Jinan Health Commission (2021-1-44)."

Response: The amended Funding Statement: “This work was supported by the Science and Technology Planning Project of Jinan Health Commission (2021-1-44). There was no additional external funding received for this study.”

Response: We have added the following data availability statement to the revised manuscript: “Data Availability Statement: No datasets were generated or analysed during the current study. All relevant data from this study will be made available upon study completion.”.

Response: Thank you for your reminder. We have updated the ORCID iD information when resubmitting the manuscript.

The point-by-point responses to the reviewers’ comments are as follows:

Reviewer #1:

Thank you very much for your comments and suggestions.

1. Question: (line 15,16,17- investigate the application of concentrated growth factor (CGF) in mandibular third molar extraction. Specifically, we aim to)

Response: As suggested by the reviewer, we revised our manuscript. We have deleted this sentence in our revised manuscript (Page 1, lines 12, 13 and 14).

2. Question: (line 27- bias risk assessment-What tool?)

Response: Thank you for your question. In response to this concern, we have added detailed information to the revised manuscript (Page 2, line 24).

3. Question: (line 61-Some studies- You said some studies but there is only one reference. Are there more studies?)

Response: Thank you for this careful review. As you indicated, the expression may not be suitable. We changed the expression in the revised manuscript. We have modified “Some studies have” to “Previous research has” in our revised manuscript (Page 3, line 57).

4. Question:(line 80- [16]-Use the reference more recent: Page MJ, McKenzie JE, Bossuyt PM, Boutron I, Hoffmann TC, Mulrow CD et al. The PRISMA 2020 statement: an updated guideline for reporting systematic reviews. BMJ. 2021 Mar 29;372:n71. doi: 10.1136/bmj.n71.)

Response: Thank you for your valuable suggestion. Reference 16 has been updated to the recent reference in the revised manuscript, which can help to improve the rigor of the manuscript (Page 4, line 77). 

5. Question: (line 95-VIP Databases to identify potentially relevant articles-Please, write a sentence about grey literature search.)

Response: Thank you for your suggestion. We have described the approach used to search the grey literature in the revised manuscript (Page 5, lines 93 and 94). ProQuest, Google Scholar, and Open Grey will be searched for grey literature.

6. Question: (line 97-terms (MeSH)-In Embase database, the emtree terms should be used in the search strategy.)

Response: Thank you for pointing out this important question. Following your suggestion, we have edited this part of the manuscript (Page 5, lines 96, 97).

7. Question: (line 103-Stusy selection-I suggest the reviewers make a calibration about the selection process to evaluate the inter-evaluator concordance.)

Response: Thank you for your suggestion. We have included a PRISMA flow diagram, which includes the additional information on the article retrieval, screening, and inclusion process.

8. Question: (line 132-interventions- Information about specific blood and growth factor extraction in each study is important extracted.)

Response: Thank you for your valuable suggestion. We have described the “interventions” in detail in the revised manuscript (Page 7, lines 133 and 134). 

9. Question: (line 139-Risk of bias assessment-There is a new version of the Cochrane risk-of bias tool: ROB2. In this tool, the studies are assessed according to each outcome.)

Response: Thank you for your excellent question. As suggested by the reviewer, we revised our manuscript. We applied the Cochrane risk-of bias tool: ROB2 in the manuscript (Page 7, lines 141-146). 

10. Question: (line 161-Subgroup analysis-The subgroup analysis about the time of follow-up might be interesting for inflammatory outcomes.)

Response: As suggested by the reviewer, we have made this modification according to your suggestion (Page 8, line 165). 

11. Question: (line 191,192,193-has/This review will only include English and Chinese studies, -But this is a protocol. Was the study not made yet, was it?)

Response: Thank you for your question. The manuscript is a protocol. However, the limitations objectively exist, so we used the present tense. We acknowledge that articles in all languages should be included. However, due to resource restrictions and the non-availability of translations, we had to restrict to those in the English and Chinese languages.

12. Question: (line 201,202-S1 Checklist. PRISMA-P (Preferred Reporting Items for Systematic review and Meta202 Analysis Protocols) 2015 checklist. (DOC)-Use the new version of the PRISMA checklist.)

Response: Thank you for bringing up this issue. The “PRISMA 2020 Checklist” is the new version of the PRISMA checklist (http://www.prisma-statement.org/PRISMAStatement/). However, there is no new version for PRISMA-P (http://www.prisma-statement.org/Extensions/Protocols.aspx). The manuscript is a protocol for meta-analysis. Therefore, we applied the PRISMA-P 2015 checklist.

13. Question: (line 278,279,280-[16] D. Moher, A. Liberati, J. Tetzlaff, D.G. Altman, P.G. The, Preferred Reporting Items for Systematic Reviews and Meta-Analyses: The PRISMA Statement, PLOS Medicine 6(7) (2009) e1000097.)-there is a new version of the PRISMA guideline.)

Response: Thank you for pointing this out. Reference 16 has been updated to a recent reference in the revised manuscript.

Reviewer #2:

Thank you very much for your comments and suggestions.

1. Question: (1- 1.1 Please insert the phrase "We performed the systematic review..." in the future, as the authors have not yet finalized the review and this article is a protocol;)

Response: Thank you for your suggestion. We have made the corresponding changes in the revised manuscript (Pages 1, lines 12).

2. Question: (1.2 Were searches carried out in the gray literature, such as Google Scholar, ProQuest, OpenGray? Has the reference list of the included articles been checked?)

Response: Thank you for bringing this issue to our attention. We will search the grey literature through ProQuest, Google Scholar, and Open Grey (Page 5, lines 93 and 94). We will also identify potentially relevant studies by reviewing the references of the included studies (Page 6, lines 110 and 111).

3. Question: (2.1 I think the introduction is too long. I recommend shortening the first paragraph a little;)

Response: Thank you for your professional question. In accordance with your suggestion, we have carefully revised the manuscript to simplify the English and to make it clearer and easier to read (Page 2, lines 33-36; Page 3, lines 43 and 44, lines 49-51). 

4. Question: (2.2 In the sentence "Because more articles have recently been published on this subject, we conducted this meta-analysis", change the term meta-analysis to systematic review.)

Response: Thank you for this careful review. We revised our manuscript. The corresponding changes have been made in the revised manuscript (Page 4, line 68). 

5. Question: (3.1 The way the PICO strategy was described was confusing. I suggest a clearer standardization;)

Response: In accordance with your suggestion, we have corrected this in our revised manuscript (Page 4, lines 80-84). 

6. Question: (3.2 Was there a date limitation when conducting the search?)

Response: Yes. Only studies published up to May 31, 2024 were considered (Page 5, line 99). 

7. Question: (3.3 I suggest that the authors use the version of PRISMA in which they have the option of adding the results of the gray literature. I believe that more studies could be found and included in the review;)

Response: Your suggestions are important. As you indicated, the grey literature is important. We have described the approach used to search the grey literature in the revised manuscript (Page 5, lines 93 and 94). 

8. Question: (3.4 As one of the aims of publishing a systematic review protocol is to allow other researchers to access it and be guided in the development of their protocols, I believe that adding the search strategy for other databases, in addition to PUBMED, is relevant, even as a supplementary file;)

Response: We are very grateful for the reviewer's comment, which helped us to improve the credibility and repeatability of our research. We have provided detailed search strategies for all databases in S1 Appendix (Page 5, lines 100 and 101). 

9 Question: (4.1 In this sentence: "Many studies have proven the efficacy of FBC in the extraction of lower third molars; however, there is no consensus", make it clear whether these are RCTs or another study design;)

Response: Many thanks for your professional comments. We revised this sentence in the manuscript to make it clearer for the reader (Page 19, lines 187-189). 

We have tried our best to revise and improve the manuscript and have made substantial changes to the manuscript according to the reviewers’ comments.

We look forward to your feedback about the revised manuscript, and we thank you for your comments.

Thank you very much for your work on this paper.

We wish you all the best!

Sincerely,

Jin Wang

---

## [Decision Letter · Decision Letter 1]

9 Apr 2024

Application of Concentrated Growth Factor in Mandibular Third Molar Extraction: A Protocol for Systematic Review and Meta-Analysis

PONE-D-24-01317R1

Dear Dr. Wang,

We’re pleased to inform you that your manuscript has been judged scientifically suitable for publication and will be formally accepted for publication once it meets all outstanding technical requirements.

Kind regards,

Endi Lanza Galvão

Academic Editor

PLOS ONE

Additional Editor Comments (optional):

Reviewers' comments:

Reviewer's Responses to Questions

**Comments to the Author**

1. Does the manuscript provide a valid rationale for the proposed study, with clearly identified and justified research questions?

Reviewer #1: Yes

Reviewer #2: Yes

2. Is the protocol technically sound and planned in a manner that will lead to a meaningful outcome and allow testing the stated hypotheses?

Reviewer #1: Yes

Reviewer #2: Yes

3. Is the methodology feasible and described in sufficient detail to allow the work to be replicable?

Reviewer #1: Yes

Reviewer #2: Yes

4. Have the authors described where all data underlying the findings will be made available when the study is complete?

Reviewer #1: Yes

Reviewer #2: Yes

5. Is the manuscript presented in an intelligible fashion and written in standard English?

Reviewer #1: Yes

Reviewer #2: Yes

6. Review Comments to the Author

You may also provide optional suggestions and comments to authors that they might find helpful in planning their study.

Reviewer #1: Thank you for responding to or justifying all suggested changes. The protocol is clearer and with due methodological rigor and therefore deserves to be published.

Reviewer #2: None comments.

7. PLOS authors have the option to publish the peer review history of their article (what does this mean?). If published, this will include your full peer review and any attached files.

Reviewer #1: No

Reviewer #2: No
